# Archetypes of Goal and Scope Definitions for Consistent Allocation in LCA

**Dieuwertje Schrijvers** **, Philippe Loubet and Guido Sonnemann ***

Institute of Molecular Sciences (ISM), University of Bordeaux, CNRS, Bordeaux INP, UMR 5255, F-33400 Talence, France; dieuwertje.schrijvers@u-bordeaux.fr (D.S.); philippe.loubet@enscbp.fr (P.L.)
* Correspondence: guido.sonnemann@u-bordeaux.fr

**Abstract:** The selection of an appropriate allocation procedure for co-production and recycling in Life Cycle Assessment (LCA) depends on the goal and scope of the analysis. However, it is not always clear when partitioning or system expansion can be applied, or when to conduct an attributional or a consequential LCA, both for LCA practitioners and users of LCA results. In this paper, the influence of the goal and scope on the selected modeling approaches is clarified. The distinction between process-oriented and product-oriented LCAs, between system expansion and substitution, and between the cut-off approach and other allocation procedures are highlighted. Archetypes of goal and scope definitions are developed. These archetypes reflect the minimum amount of information required to select an allocation procedure. It is demonstrated via an illustrative example that the question "what is the environmental impact of a product" can result in at least 15 different research questions requiring at least five different modeling methods. Finally, perspectives are provided on the use of attributional and consequential approaches to evaluate the environmental, social, and economic sustainability of products and processes.

**Keywords:** life cycle assessment; allocation; recycling; attributional LCA; consequential LCA; sustainability; goal and scope; life cycle inventory

## 1. Introduction

When conducting a Life Cycle Assessment (LCA), an allocation procedure is often required due to joint co-production or recycling within the system under study, which makes the system multifunctional. Many LCA practitioners find applying allocation challenging, as various allocation procedures are available, guidelines provide divergent recommendations, and all allocation procedures seem to be in line with the International Organization for Standardization (ISO) standard 14044 [1]. Allocation can have a significant influence on the LCA results, which makes the selection of an allocation procedure an important step within the LCA.

ISO 14044 describes a preference for system expansion when a multifunctional situation must be handled [2]. This is often interpreted as "substitution", i.e., the modeling of the effects of co-production or recycling by the substitution of alternative processes. Other names for this modeling technique are the "avoided burden" or "end-of-life recycling" method [3]. Substitution is recommended by many guidance documents for the modeling of co-production and recycling [1]. Heijungs [4] argues against the use of system expansion as a synonym for substitution, as system expansion is described in ISO 14044 as the inclusion of the additional functions related to the co-products. This could be interpreted as a requirement to redefine the functional unit, whereas substitution is often applied by subtracting "avoided processes" from the system under study.

An alternative approach to modeling a multifunctional situation is partitioning. This method, often simply referred to as "allocation", represents the split of the life cycle inventory following a

partitioning criterion such as the relative economic revenue or the relative mass of the co-products. Several guidelines recommend to apply partitioning as an alternative for substitution for the modeling of co-products [1]. Partitioning is rarely used for recycled products. Instead, often, the cut-off approach—in which recycled materials are burden-free—is suggested [1].

According to the Global Guidance Principles for Life Cycle Assessment Databases of the UNEP/SETAC Life Cycle Initiative [5], the choice between partitioning and substitution depends on the LCA approach that is applied: partitioning is generally applied in attributional LCAs (ALCAs), and substitution in consequential LCAs (CLCAs). Following these developments, Schrijvers et al. [4] proposed a framework that assigned the partitioning method, system expansion, and the cut-off approach to ALCA, and substitution to CLCA. However, ISO 14044 does not make this distinction between ALCA and CLCA. Instead, it is recommended to conduct a sensitivity analysis when multiple allocation procedures seem applicable [2]. The International Reference Life Cycle Data System (ILCD) Handbook developed by the European Joint Research Centre (JRC) provides recommendations on when to use ALCA and CLCA, based on whether the LCA aims to provide micro-level or meso-macro-level decision support, respectively [6]. This approach was criticized by Ekvall et al. [7], who referred to the notion that CLCA should always be used as a decision-support tool. In practice, most LCA practitioners know that the chosen allocation procedure is dependent on the goal and scope of their study. However, it is less well understood when an ALCA or CLCA is appropriate, nor how to choose, for example, between system expansion and partitioning.

LCA results are now also increasingly used beyond the strict domain of LCA. For example, in the research area of raw material criticality—which evaluates the probability of the disruption of the supply of a raw material and the vulnerability of a company, technology, or economy to such a disruption—LCA results are used to identify whether environmental implications could result in a disrupted supply of a raw material or whether raw material use is environmentally sustainable [8–12]. Sonnemann et al. [13] aimed to clarify the link between LCA and criticality assessments within a Life Cycle Sustainability Assessment (LCSA) framework, which evaluates the environmental, economic, and social sustainability of a product. In their proposed framework, environmental LCA is used to evaluate a product's environmental sustainability, whereas criticality indicators—that evaluate the potential of a supply disruption and the vulnerability to such a disruption—are used to evaluate economic sustainability. However, the distinction between ALCA or CLCA results has not been addressed—neither within the cited criticality assessments, nor within the LCSA framework.

This paper aims to provide further guidance in the choice of a suitable allocation procedure based on the goal and scope of the LCA study. Furthermore, it is discussed what types of information ALCAs and CLCAs provide in the context of a sustainability assessment. In this regard, the paper is constructed as follows. In the "Materials and Methods" section, elements of the goal and scope of the LCA study are presented that are relevant in the formulation of research questions that allow for the identification of a suitable allocation procedure. In the Results section, allocation procedures are connected to these elements of the goal and scope in a revised framework for goal-dependent allocation. The aforementioned research questions are presented in the form of archetypes of goal and scope definitions. In the Discussion section, the implications of the framework and the archetypes are discussed with regard to the use of the wording "system expansion" and "substitution", and the relevance of the cut-off approach. Furthermore, results are discussed for use in sustainability assessments, in particular for economic sustainability. Conclusions are provided in Section 5.

## 2. Materials and Methods

In this section, we present the elements of the goal and scope that influence the choice of allocation and the modeling of the Life Cycle Inventory (LCI) as specified in this paper. This section does not provide an exhaustive overview of the LCA goal and scope, as certain aspects, such as data quality requirements, do not affect the allocation strategy. Whereas the approach put forward in this paper is generally applicable to multifunctional LCA case studies, we illustrate the approach with the example

of the recycling of rare earth elements from end-of-life fluorescent lamps. This recycling process serves as a rich illustrative example, as the recycling process is relatively complex due to its multiple functional flows (as defined in [14]); it is a joint process that provides a waste treatment function and supplies multiple co-products. Fluorescent lamps are coated with phosphors, which contain, among others, the rare earth elements yttrium, europium, lanthanum, cerium, and terbium. Normally, after the recycling of glass and metals from the lamps, the remaining glass with the phosphor coating is landfilled [15]. Solvay implemented between 2012 and 2016, when the prices of rare earth elements were booming, a recycling process that extracted the original rare earth elements from the phosphorous powder, in order to make recycled phosphors [16,17]. The recycling process is schematically represented by Figure 1, which presents data obtained from Solvay [18]. The phosphorous powder is first pretreated to remove the remaining glass. Then, the rare earth elements are separated in subsequent separation reactors. In this illustrative example, we will focus on the cradle-to-gate inventory of recycled yttrium. Multifunctionality, due to the fact that this product has been recycled and due to co-production in upstream processes, can be solved at this stage, before a complete cradle-to-grave LCA study is conducted.

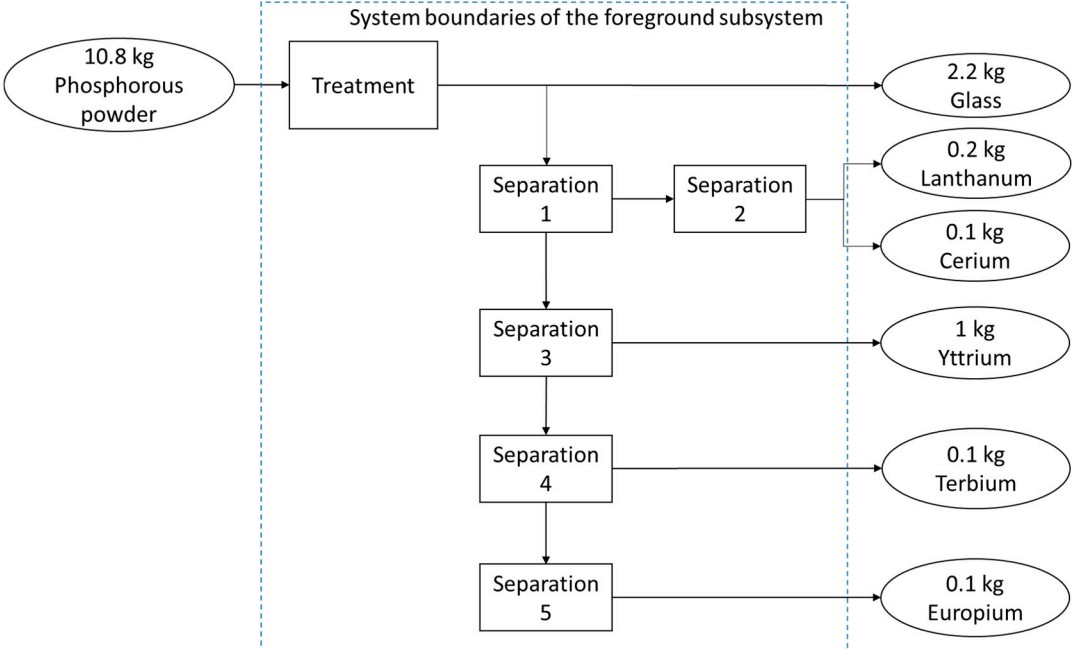

**Figure 1.** Schematic overview of the recycling process of yttrium from phosphorous powder. The foreground subsystem contains processes that are directly affected by the study [19], i.e., the processes operated by the company Solvay.

## 2.1. The Reason for Carrying Out the Study

When one is interested in the cradle-to-gate environmental impacts of recycled yttrium, this could be motivated by two main reasons. First of all, the LCA practitioner might wonder whether the recycling of yttrium from phosphorous powder is environmentally beneficial over the non-recycling of yttrium, or perhaps compared to a different recycling route for yttrium. This LCA practitioner might also be interested in finding the environmental hotspots within the recycling process, in order to improve the recycling activity. In this case, the LCA practitioner has a pronounced interest in optimizing the process of obtaining yttrium, rather than the use of yttrium in a product. Hence, the person can conduct a process-oriented LCA. Alternatively, an LCA practitioner might want to use recycled yttrium in his/her product and needs to know the environmental impacts related to having recycled yttrium as an input in his/her system, in other words, the environmental impacts related to the consumption of recycled yttrium. In that case, he/she will conduct a product-oriented LCA.

The distinction between a process-oriented LCA and a product-oriented LCA helps to provide more detail about the subject of the LCA. It sheds light on the reason why the LCA practitioner is interested in the collection of certain environmental information.

*2.2. The Perspective of the LCA Practitioner*

The perspective of the intended audience of the study results, and by extension of the LCA practitioner in the specific LCA study, determines whether an attributional or a consequential approach should be applied. An ALCA aims to evaluate the environmental impacts that can directly be associated with the consumption or the production of recycled yttrium. Hence, the LCA practitioner identifies which products were used and which processes took place in order to produce recycled yttrium. It has been suggested that ALCAs can be used to apply environmental taxation or to identify "who is to carry the burden" [20]. Hence, an ALCA is conducted to identify for which environmental impacts recycled yttrium is accountable.

A consequential approach is appropriate when the LCA practitioner aims to identify the environmental consequences of the consumption or the production of recycled yttrium. The LCI is constructed by first answering the question "Which processes are affected by increasing or decreasing the consumption or production of a certain product?" The increased or decreased consumption or production of recycled yttrium can result in changes in the consumption and production of products in other product value chains, due to economic causal relationships. Hence, the consequential analysis is not limited to the impacts of the activities that take place within a product's value chain, but considers activities that are affected on a global level.

*2.3. The Functional Unit*

The initial functional unit of a cradle-to-gate process-oriented LCA with a focus on recycled yttrium could be "the production of 1 kg of recycled yttrium", and of the cradle-to-gate product-oriented LCA "the consumption of 1 kg of recycled yttrium". If the product system contains multifunctional processes, this functional unit could be adapted based on the selected allocation procedure, as discussed in Section 3.1. Of course, other information regarding the properties of the material that affect its function can be specified as well.

*2.4. The Intended Application of the Results*

Three areas of applications can be identified that would influence how the LCA study is framed and how the results are presented [21]:

- Quantifying the impacts of a product or service, in order to use these values in another LCA study in which this product or service is used. Furthermore, quantified impacts can be used for the communication of environmental impacts, for example via environmental labeling, or periodic monitoring.
- Identifying opportunities to improve the environmental performance of the product or service, for example via innovation. For this type of application, a contribution analysis, or hotspot analysis aids in identifying the products and processes that contribute most to the environmental impacts of the functional unit.
- Making a decision to select between multiple alternative options. In the end, most LCA studies have the purpose to compare two alternative products or services that provide the same function. Furthermore, it is interesting to compare the impacts of a product before and after the application of an adjustment that is supposed to reduce the products' environmental impacts.

*2.5. Relationship between the Elements of the Goal and Scope*

The four elements of the goal and scope presented above are sometimes considered as interlinked. An LCA is often conducted with multiple applications in mind. First, a hotspot analysis could be

done to identify which processes within the life cycle have a large contribution to environmental impacts. Then, a comparative LCA could show whether an improvement strategy indeed results in lower environmental impacts of a certain process. The results of such a comparison could be used to support the decision on implementing the improvement action or not. Other authors have argued that such decision support always requires a consequential LCA [22], because the modifications that are done within the process under study could result in indirect impacts, or even environmental benefits, in other product value chains. It is true that such a complete overview of environmental impacts and benefits is not obtained via an ALCA, as the latter only evaluates environmental impacts that are directly associated with a product's value chain. However, the results of an ALCA can still support decision-making when only the value-chain impacts are of concern to the LCA practitioner, a perspective that is further discussed in Section 4.2. Hence, the choice between an ALCA or CLCA is determined by the perspective of the LCA practitioner, and not, as suggested elsewhere [7,22], by the application of the results as categorized in Section 2.4. Therefore, the elements of the goal and scope as presented above can be defined independently of one another.

## 3. Results

### 3.1. Updated Framework for Goal-Dependent Allocation

Based on the reason for carrying out the study and the perspective of the LCA practitioner, a suitable allocation procedure can be selected. The dependency of the allocation procedure on these elements of the goal and scope is shown in Figure 2, which is an updated version of the framework developed by Schrijvers et al. [3]: Figure 2 includes the process-oriented and product-oriented perspectives, a link between CLCA and system expansion, and excludes the cut-off approach, as further discussed in Section 4.

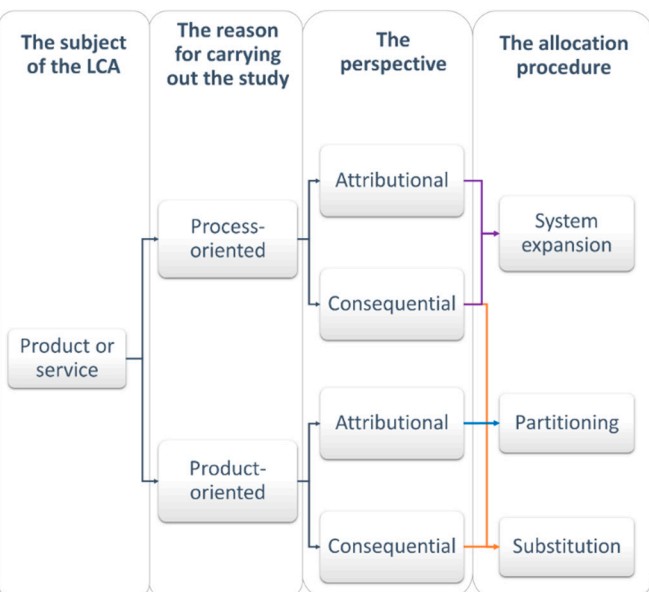

**Figure 2.** Dependency of the allocation procedure on the goal and scope of the Life Cycle Assessment (LCA).

Figure 1 shows that the recycling process of yttrium is multifunctional due to the fact that a waste flow is treated, and several co-products are produced. In a process-oriented LCA on the production of recycled yttrium, the focus of the study is not necessarily put on yttrium, but rather on the recycling process itself. Therefore, allocation can be avoided by applying system expansion. The functional unit can be expanded to include the additional functions of the recycling process: "the treatment of 10.8 kg

of phosphorous powder, and the production of 1 kg of yttrium, 2.2 kg of glass, 0.2 kg of lanthanum, 0.1 kg of cerium, 0.1 kg of terbium, and 0.1 kg of europium".

However, in a product-oriented LCA, an allocation procedure is required to identify the impacts of the consumption of only 1 kg of yttrium. Partitioning can be applied to identify which share of the environmental impacts of the recycling process and other upstream processes related to the supply of the phosphorous powder is attributable to the recycled yttrium in an attributional product-oriented LCA. In a consequential product-oriented LCA, the effects of the production or the consumption of co-products, recycled products, or wastes are modeled by substitution [3,5]. Substitution can also be applied to model the effects of the consumption of phosphorous powder and the production of glass, lanthanum, cerium, terbium, and europium in a consequential process-oriented LCA with the functional unit "the production of 1 kg of recycled yttrium". Whereas system expansion avoids the need to model these flows by substitution, knowledge on the effects of these other functional flows on the environment can be valuable.

The distinction between process-oriented and product-oriented LCAs was already made by Azapagic & Clift [19], although it is currently not often applied. This distinction clarifies that system expansion can only be applied in a process-oriented LCA. This is in line with the allocation hierarchy of ISO 14044 that states that "wherever possible, allocation should be avoided by expanding the product system to include the additional functions related to the co-products [ ... ]" [2]. In other words, allocation can be avoided by conducting a process-oriented LCA, which provides information about the process itself and not about the individual inputs (for joint waste treatment or recycling processes) or outputs (in the case of joint co-production).

*3.2. Archetypes of Goal and Scope Definitions*

Based on the four elements of the goal and scope presented in Section 2, we can construct archetypes of goal and scope definitions. These archetypes are presented in the form of research questions that provide a sufficient amount of detail to enable the identification of a suitable allocation procedure.

Table 1 presents the parameters of interest of the LCA practitioner, which are integrated into the questions formulated in Table 2. Each Greek symbol reflects two possible parameter values—i.e., α represents either the ingoing (treatment) and/or outgoing (production) flows of a process, or the consumption of a product, β represents either the accountability for impacts, or consequences on global impacts, and γ reflects either only the subject of the LCA or the subject of the LCA as well as additional functions provided by the other functional flows. The selection of the parameter values indicates the appropriate LCA modeling approach via Figure 2. For example, if the LCA practitioner is interested in the life cycle impacts for which the user of recycled yttrium can be held accountable, the parameter values for α, β, γ are "the consumption", "accountability for impacts", and "the subject of the LCA" (here: recycled yttrium), respectively. Filling these parameter values into the first question of Table 2 results in the question "What is the accountability for impacts of the consumption of recycled yttrium?" The resulting LCA modeling approach is an attributional product-oriented LCA, using the allocation procedure "partitioning", as indicated by Figure 2. Table 3 shows all potential research questions that can be formulated based on the initial subject of the LCA of "recycled yttrium". The system boundaries corresponding to these questions are represented by Figures 3–5. From Table 3 it becomes evident that a single subject can already lead to 15 different research questions, with at least five different combinations of LCA approaches and modeling specifics. These five modeling approaches are of general relevance, because they are applicable to each LCA case study with at least two functional flows (a waste treatment service and/or the production of a (co-)product, see [14])—i.e., each multifunctional problem in LCA.

**Table 1.** Building blocks of archetypes of LCA goal and scope definitions. The parameters of interest represented by α, β, and γ are used in the archetypical goal and scope definitions as defined in Table 2.

| Item of the Goal and Scope Definition | Parameters of Interest | Integration into Table 2 | LCA Modeling Approach |
|---|---|---|---|
| The reason for carrying out the study | The production/treatment The consumption | α | Process-oriented LCA Product-oriented LCA |
| The perspective of the LCA | Accountability for impacts Consequences on global impacts | β | Attributional LCA Consequential LCA |
| The functional unit | The subject of the LCA The subject of the LCA and additional functions | γ | Partitioning/Substitution System expansion |

**Table 2.** Archetypical LCA goal and scope definitions. Parameter values are presented in Table 1.

| Intended Application of the Results | Research Question |
|---|---|
| Quantifying environmental impacts Identifying opportunities for improvement Decision-making | What is/are the β of α of γ? How can we decrease the β of α of γ? Does α of γ have (a) lower β than its alternative? |

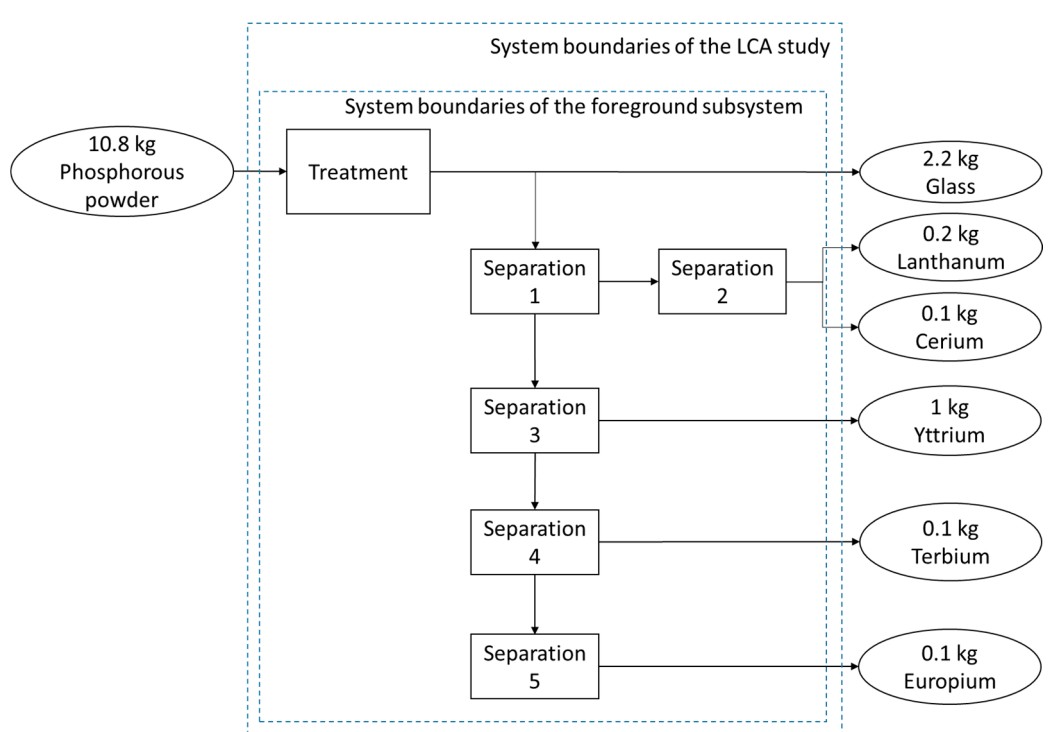

**Figure 3.** System boundaries of an attributional or consequential process-oriented LCA with the functional unit "the treatment of 10.8 kg of phosphorous powder, and the production of 1 kg of yttrium, 2.2 kg of glass, 0.2 kg of lanthanum, 0.1 kg of cerium, 0.1 kg of terbium, and 0.1 kg of europium".

**Table 3.** Potential research questions and corresponding LCA approaches and system boundaries based on the initial subject of "recycled yttrium", using the archetypical LCA goal and scope definitions of Table 2. For the CLCAs, it is assumed that the modeled change is marginal.

| LCA Approach | Functional Unit | Research Question | Modeling Specifics |
|---|---|---|---|
| Attributional process-oriented LCA | The treatment of 10.8 kg of phosphorous powder, and the production of 1 kg of yttrium, 2.2 kg of glass, 0.2 kg of lanthanum, 0.1 kg of cerium, 0.1 kg of terbium, and 0.1 kg of europium | 1. What is the accountability for impacts of the treatment of 10.8 kg of phosphorous powder, and the production of 1 kg of yttrium, 2.2 kg of glass, 0.2 kg of lanthanum, 0.1 kg of cerium, 0.1 kg of terbium, and 0.1 kg of europium via the recycling process? <br> 2. How can we decrease the accountability for impacts of the treatment of 10.8 kg of phosphorous powder, and the production of 1 kg of yttrium, 2.2 kg of glass, 0.2 kg of lanthanum, 0.1 kg of cerium, 0.1 kg of terbium, and 0.1 kg of europium via the recycling process? <br> 3. Does the treatment of 10.8 kg of phosphorous powder, and the production of 1 kg of yttrium, 2.2 kg of glass, 0.2 kg of lanthanum, 0.1 kg of cerium, 0.1 kg of terbium, and 0.1 kg of europium via the recycling process have a lower accountability for impacts than the treatment and production of these flows via alternative processes? | Only the LCI of the foreground subsystem is calculated based on attributional background data. No allocation is necessary within the foreground subsystem, as system expansion is applied (Figure 3). |
| Attributional product-oriented LCA | The consumption of 1 kg of recycled yttrium | 4. What is the accountability for impacts of the consumption of 1 kg of recycled yttrium? <br> 5. How can we decrease the accountability for impacts of the consumption of 1 kg of recycled yttrium? <br> 6. Does the consumption of 1 kg of recycled yttrium have a lower accountability for impacts than the consumption of 1 kg of primary yttrium? | Partitioning must be applied to identify the cradle-to-gate inventory that is attributed to recycled yttrium. More information is required from the product system that supplies the phosphorous powder, as the recycling process is part of this product system (Figure 4). |
| Consequential process-oriented LCA | The treatment of 10.8 kg of phosphorous powder, and the production of 1 kg of yttrium, 2.2 kg of glass, 0.2 kg of lanthanum, 0.1 kg of cerium, 0.1 kg of terbium, and 0.1 kg of europium | 7. What are the consequences on global impacts of the treatment of 10.8 kg of phosphorous powder, and the production of 1 kg of yttrium, 2.2 kg of glass, 0.2 kg of lanthanum, 0.1 kg of cerium, 0.1 kg of terbium, and 0.1 kg of europium via the recycling process? <br> 8. How can we decrease the consequences on global impacts of the treatment of 10.8 kg of phosphorous powder, and the production of 1 kg of yttrium, 2.2 kg of glass, 0.2 kg of lanthanum, 0.1 kg of cerium, 0.1 kg of terbium, and 0.1 kg of europium via the recycling process? <br> 9. Does the treatment of 10.8 kg of phosphorous powder, and the production of 1 kg of yttrium, 2.2 kg of glass, 0.2 kg of lanthanum, 0.1 kg of cerium, 0.1 kg of terbium, and 0.1 kg of europium via the recycling process have lower consequences on global impacts than the treatment and production of these flows via alternative processes? | Only the LCI of the foreground subsystem is calculated based on consequential background data. No allocation is necessary within the foreground subsystem, as system expansion is applied (Figure 3). |

**Table 3.** *Cont.*

| LCA Approach | Functional Unit | Research Question | Modeling Specifics |
|---|---|---|---|
| Consequential process-oriented LCA | The production of 1 kg of recycled yttrium | 10. What are the consequences on global impacts of the production of 1 kg of recycled yttrium?<br>11. How can we decrease the consequences on global impacts of the production of 1 kg of recycled yttrium?<br>12. Does the production of 1 kg of recycled yttrium have lower consequences on global impacts than the production of 1 kg of primary yttrium? | The LCI of the foreground subsystem is calculated based on consequential background data. The ingoing flow of phosphorous powder is modeled by the substitution of its marginal waste treatment process. The outgoing flows of glass, lanthanum, cerium, terbium, and europium are modeled by the substitution by their marginal users (Figure 4). |
| Consequential product-oriented LCA | The consumption of 1 kg of recycled yttrium | 13. What are the consequences on global impacts of the consumption of 1 kg of recycled yttrium?<br>14. How can we decrease the consequences on global impacts of the consumption of 1 kg of recycled yttrium?<br>15. Does the consumption of 1 kg of recycled yttrium have lower consequences on global impacts than the consumption of 1 kg of primary yttrium? | If the supply of recycled yttrium is constrained, the LCI represents the substitution of recycled yttrium by its marginal user. If the supply is unconstrained, the LCI of the foreground subsystem is calculated. The ingoing flow of phosphorous powder is modeled by the substitution of its marginal waste treatment process. The outgoing flows of glass, lanthanum, cerium, terbium, and europium are modeled by the substitution by their marginal users (Figure 5). |

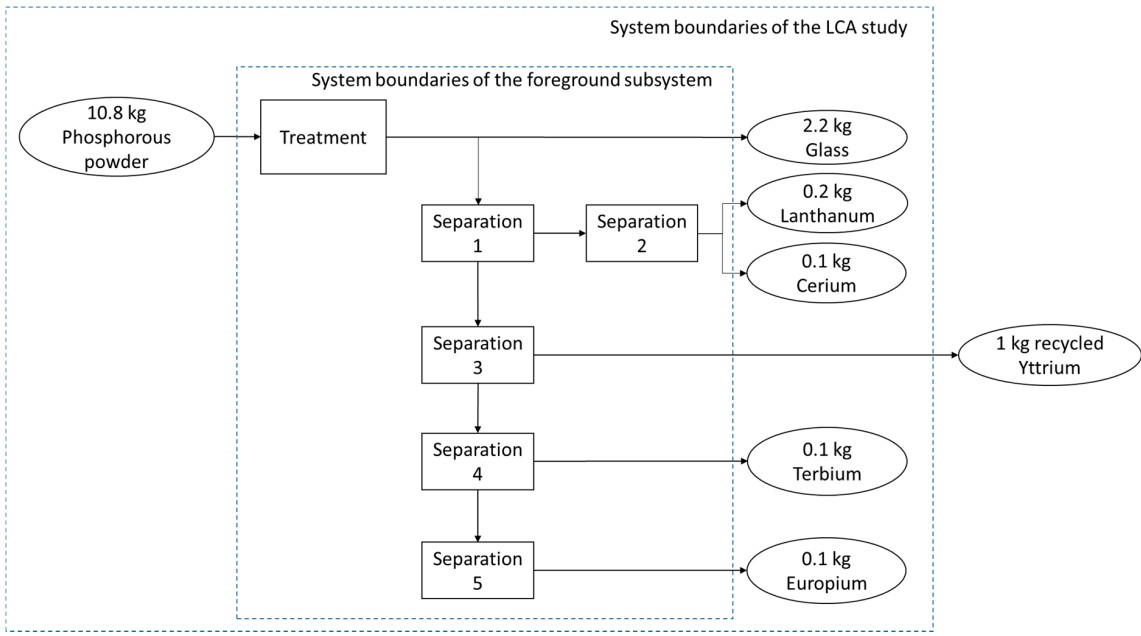

**Figure 4.** System boundaries of an attributional product-oriented LCA with the functional unit "the consumption of 1 kg of recycled yttrium", or of a consequential process-oriented LCA with the functional unit "the production of 1 kg of recycled yttrium".

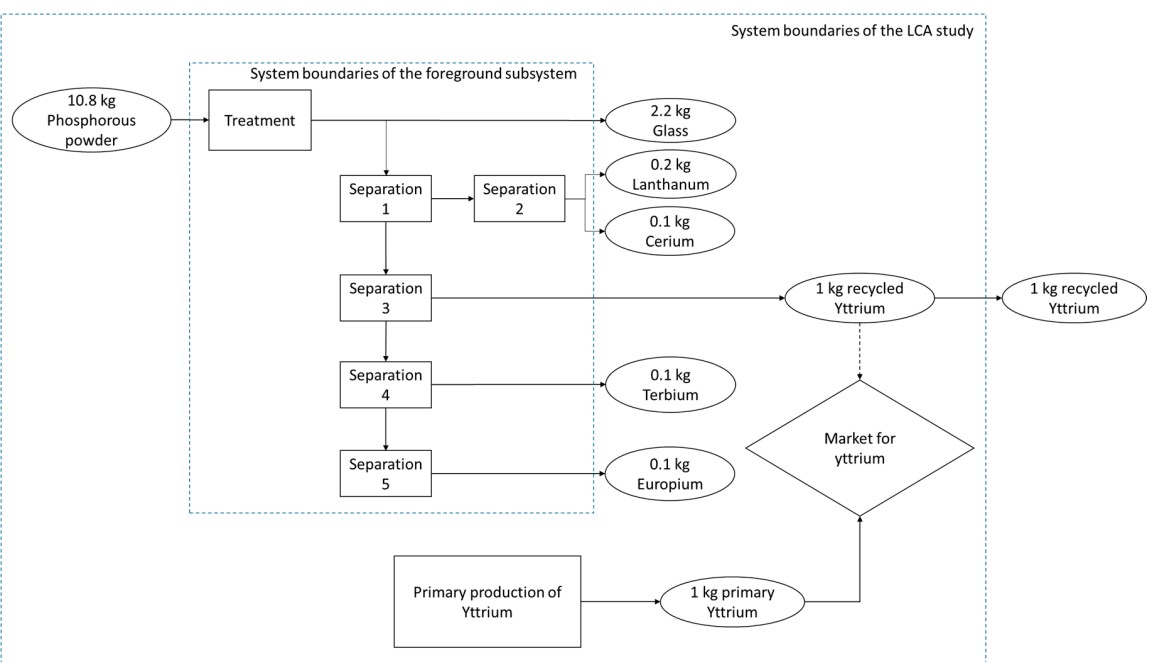

**Figure 5.** System boundaries of a consequential product-oriented LCA with the functional unit "the consumption of 1 kg of recycled yttrium", where this consumption results in an additional demand for recycled yttrium from the market.

## 4. Discussion

### 4.1. Untangling System Expansion, Substitution, and the Cut-Off Approach

Based on Figure 2 we can address several common practices that persist in the LCA domain. First of all, system expansion and substitution are not synonyms of one another, as is also argued by Heijungs [4]. System expansion can be applied both in an ALCA and a CLCA, while substitution can

only be applied in a CLCA. It is sometimes argued (e.g., [6]) that system expansion and substitution are mathematically the same, which could be the reason why many guidance documents recommend applying substitution in an ALCA [1]. According to this argument, it is not necessary to add additional functions to the functional unit, as is done in the attributional process-oriented LCA (questions 1–3 in Table 3). Instead, the functional unit remains "1 kg of recycled yttrium". The LCI of alternative production and treatment routes of the co-functions is subtracted from the LCI of the foreground process of Figure 1. This leads to four problems:

1.  Questions 1 and 2 of Table 3 cannot be answered via the operation of subtracting functions instead of the addition of functions.
2.  Equivalence between system expansion and substitution is only achieved for question 3, and only with regard to the absolute difference in environmental impacts (e.g., "the alternative system is accountable for 20 kg $CO_2$-eq more than the system under study"). Any other comparison between the two product systems, such as percentages of improvements, is distorted due to the subtraction of functions instead of the addition of functions.
3.  The results can be easily misinterpreted by the suggestion that the alternative production routes are substituted, while in an attributional process-oriented LCA, a comparison between two product systems can be done for the mere purpose of benchmarking.
4.  Keeping the functional unit as "1 kg of recycled yttrium" suggests that a product-oriented LCA is conducted. Results are easily used for downstream studies in which yttrium is used, while the results do not refer to the cradle-to-gate impacts attributed to yttrium, as the applied approach is a process-oriented LCA.

Due to the limitations of using system expansion and substitution as synonyms, we encourage the use of the terms only according to the following definitions (based on ISO 14044 [2]):

*   System expansion: Including the additional functions related to the co-products in the functional unit.
*   Substitution: Allocation of inputs and outputs between the products and functions in a way that reflects economic relationships.

Another common practice is the use of the cut-off approach as a distinct allocation procedure. The cut-off approach considers the recycled material that is used in a system as burden-free, and several guidelines recommend this approach [1]. Furthermore, the background LCI database ecoinvent proposes a system model in which the cut-off approach is applied to recycled materials [23]. It should be noted that ISO 14044 does not mention the cut-off approach as an allocation procedure. Instead, the standard allows for the definition of cut-off criteria that determine which inputs and outputs can be excluded from the analysis, e.g., based on mass, energy, or environmental significance [2]. From the modeling approaches suggested in Table 3, three strategies could be identified that could be understood as a cut-off approach which are in line with the establishment of cut-off criteria according to ISO 14044:

1.  In an attributional product-oriented LCA, when partitioning is applied, it could appear that the co-product or recycled product has a very low mass (for mass allocation) or economic revenue (for economic allocation) compared to the other co-products of the system. In that case, the allocation factor for the low-value product could approach, or be, 0%. With an allocation factor of 0%, the product becomes burden free, which is in line with the cut-off approach [3]. However, the cut-off approach is not appropriate when the allocation factor is low in an economic allocation due to the fact that the co-product, or recycled product, is produced in a low quantity, while the unit price is relatively high, as in this case the LCI per unit of product is non-negligible.
2.  In a consequential product-oriented or process-oriented LCA (questions 10–15), it is possible that the flows which are modeled by substitution generate relatively low impacts. In that case, it could be argued that the environmental consequences due to the multifunctional flow are negligible and are cut-off.

3.  Alternatively, if an LCA practitioner argues that the phosphorous powder is burden-free, because it is a waste, he/she might aim to conduct a process-oriented LCA where the flow of ingoing waste is a flow of interest. Instead of referring to this flow as "burden-free", the flow should be included in the functional unit. As shown in Figure 3, functional flows do not "carry" an impact, as the overall impact of these flows is the subject of analysis.

Recommending the cut-off approach as a distinct allocation procedure for systematic application to recycling situations implies that recycling does not have a significant value. This does not do justice to the potentially high environmental benefits of recycling, especially considering the political and industrial efforts of moving towards a circular economy. Therefore, we would argue for using the cut-off approach only as a simplification strategy in the rare cases when recycling has limited economic value and environmental benefits.

*4.2. Perspectives for the Use of Attributional and Consequential Approaches in Sustainability Assessments*

While it is already useful to know whether answering a research question requires an attributional or a consequential approach, the relevance of the research questions of Table 3, or in other words, of an attributional and a consequential approach, should be discussed.

An attributional approach can be used to evaluate environmental impacts, and similarly social circumstances, that can be associated with activities that take part of a product's value chain. For example, if a company decides not to source a material from a mine that is known to have poor waste management practices, causing high environmental impacts, these environmental impacts will not appear in the value chain of this company anymore. This company's strategy does not necessarily mean that the production volume of this mine and the corresponding waste flows are reduced. It is possible that supply chains are reconfigured in such a way that other product value chains are now more dependent on this highly polluting mine. A similar reflection may be valid for social circumstances. A company can decide not to source from conflict regions, which could be evaluated with, for example, a Life Cycle Attribute Assessment that identifies the share of a product's supply chain that has a certain social attribute [24]. A company's decision to stop sourcing from conflict regions does not guarantee a decreased output from these regions – these regions potentially supply more minerals and metals to companies that do not have a transparent supply chain. This suggests that environmental and social impacts evaluated with an attributional approach do not provide information on the overall environmental or social sustainability of a product. Nonetheless, the company of this example can be associated with lower environmental impacts and social problems and is, therefore, protected against potential reputation damage. Furthermore, the company's alternative sourcing route might be less affected by regulations or legislations, such as environmental taxations, stricter waste management regulations, or conflict minerals regulations, which might disrupt the supply of materials from this supply route. The company is, therefore, less exposed to potential supply disruptions.

Potential reputation damage and potential supply disruptions are factors that can indicate whether a company's operations are stable. Continuous operations of companies contribute to the overall economic stability of a region or a country, by the provision of goods, jobs, and taxes [25]. Therefore, potential reputation damage and supply disruptions reflect the economic sustainability of a value chain. This is in line with the economic impact pathways developed by Neugebauer et al. [26], where the company's reputation contributes to a consumer's willingness-to-pay, which is suggested as a midpoint indicator for economic sustainability in an LCSA. Furthermore, indicators that evaluate the probability of a supply disruption, such as geopolitical, geological, but also environmental and social factors, are commonly included in Raw Material Criticality Assessments [10,13,27]. Providing a sustainable flow of income to a certain company, country, or region is generally the object of interest in criticality assessments, which underlines the relevance of indicators of potential supply disruptions to evaluate the economic sustainability of a product's value chain.

Mineral resource depletion or dissipation also has a prominent role in both LCA and Raw Material Criticality Assessments [13,27,28]. The LCA safeguard subject for "mineral resources" has only recently

been clarified within the Area of Protection of "natural resources" as "the potential to make use of the value that mineral resources can hold for humans in the technosphere. The damage is quantified as the reduction or loss of this potential caused by human activity" [29]. As argued by other authors [13,30,31], mineral resource use is not an environmental issue, but rather an economic one. Therefore, a high impact caused by mineral resource use can affect economic sustainability. An attributional approach can show whether these economic damages can be associated with the value chain under study. If this is the case, for example because an upstream mining activity does not valorize scarce by-products with high technological potential, the economic sustainability of this specific value chain is again at stake. Not because this value chain is dependent on the resource itself, but rather due to potential regulations or other efforts that aim to decrease the depletion or dissipation of this resource. Consequential LCA results could represent whether the consumption of a product leads to increased resource scarcity. This reflects the economic sustainability of the value chains in which the scarce resource is used.

A consequential approach can also be used to evaluate the net impacts, or benefits, of a product on the environment, considering direct and indirect effects which go beyond a product's value chain. Therefore, the environmental impacts reflect whether the production or the use of a product is environmentally sustainable. Furthermore, for Social LCA (SLCA), a consequential evaluation can be conducted, where the social consequences of a decision are compared with the social consequences of not taking this decision [32]. Benoît Norris [33] highlights the importance of a consequential SLCA to evaluate the costs and benefits of implementing a company's solution to improve social circumstances. This suggests that a consequential SLCA can be used for the evaluation of the social pillar of an LCSA. However, Zamagni et al. [34] do not consider the attributional and consequential divide relevant for SLCA, as every SLCA should have the goal to evaluate social consequences. This statement could be interpreted as a questioning of the usefulness of an attributional SLCA approach for a social sustainability assessment, confirming the relevance of a consequential approach.

## 5. Conclusions

Choosing a suitable allocation procedure remains challenging in environmental LCA, even though there seems to be consensus that the allocation procedure depends on whether an attributional or consequential approach is applied. However, both LCA practitioners and users of results of environmental LCA, such as in the domain of raw material criticality assessment, do not always distinguish between different LCI approaches. In fact, the question "what is the environmental impact of a product?" is not detailed enough to identify an appropriate allocation procedure. Furthermore, the distinction between an approach focusing on the impacts within a value chain or global impacts appears not only relevant for the evaluation of environmental sustainability, but also for social and economic sustainability.

In this paper, it is shown how the goal and scope, and more specifically the reason to carry out the study, the perspective of the LCA practitioner, and the functional unit influence the most appropriate allocation procedure. The distinction between process-oriented and product-oriented LCAs, between system expansion and substitution, and between the cut-off approach and other allocation procedures are highlighted. Archetypes of goal and scope definitions are developed, reflecting the minimum amount of information that is required to select an allocation procedure. It is demonstrated via an illustrative example that the question "what is the environmental impact of a product" can result in at least 15 different research questions requiring at least five different modeling methods.

It is discussed how attributional and consequential LCI can be used to evaluate the environmental, social, and economic sustainability of value chains. Increased understanding of the relevance of attributional and consequential LCI modeling in sustainability assessments beyond environmental impacts will enable more effective use of LCA results in sustainable development strategies. Furthermore, it will contribute to the development of methods that use LCA results, such as criticality assessment methods. The guidance in formulating detailed research questions could aid in the increased

consistency of LCA studies, and the improved interpretation, communication, and comparability of LC(S)A results.

**Author Contributions:** Conceptualization, D.S.; Funding acquisition, G.S.; Investigation, D.S.; Methodology, D.S.; Project administration, G.S.; Supervision, G.S.; Validation, D.S., P.L. and G.S.; Visualization, D.S.; Writing—original draft, D.S.; Writing—review and editing, P.L. and G.S. All authors have read and agreed to the published version of the manuscript.

**Funding:** Part of this work has been developed within the context of the PhD of the first author, which was funded by Solvay and the French National Association for Technical Research (CIFRE Convention No. 2013/1146). The first author, furthermore, received funding from EIT Raw Materials within the projects IRTC (International Round Table on Materials Criticality) and Suscritmat (Sustainable Management of Critical Raw Materials).

**Conflicts of Interest:** The authors declare no conflict of interest. Jean-François Viot and Françoise Lartigue-Peyrou of the Solvay Research & Innovation team "Eco-design, Modeling & Simulation" in the Department of Research & Development for Sustainable Processes contributed to this work by fruitful discussions and the provision of the case study.

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
