# Peer review of "Archetypes of Goal and Scope Definitions for Consistent Allocation in LCA"

_sustainability, doi:10.3390/su12145587_

Round 1

Reviewer 1 Report

The study analysed the concepts of “goal and scope definition” and “allocation procedure” within the Life Cycle Assessment methodology (LCA). These two points are of great interest because of the LCA modelling and the use of LCA results are a function of the purpose and the methodological choices. Moreover, the authors enlarges their view considering the possible use of LCA analyses in other sustainability method approaches, such as Life Cycle Sustainability Assessment and Criticality Assessment. The authors applied their analysis on a case study regarding the recycling process of fluorescent lamp and, in particular focussing on recycled yttrium production/consumption.

Broad comments

Introduction and aims

The authors made a general overview of the different methods to solve the multifunctionality of a targeted process and of the main guidelines provided by international standards concerning LCA method. Moreover, they expanded the framework including the possible use of LCA results and the connection between LCA studies and the future use of their results.

In order to make the introduction more complete, a paragraph describing the present methods to solve the problem about which allocation procedure has to be applied (e.g. sensibility approach using different procedures) could improve the description of the topic.

Materials and methods

Authors illustrated four points to scheme the goal and scope definition in LCA modelling: reason for carrying out the study, the perspective of the LCA practitioner, the functional unit, the intended application of the results.

In general, the points are well explained. However, these are complex points that could have several connections not only within them, but also between them. For example, process vs product-oriented LCAs and attributional vs consequential LCAs have different points of connection (e.g. the optimisation of a process – process-oriented – required a consequential LCA, since the consequences of the modifications implemented in the process for its optimisation have to be modelled and used for assessing the environmental impact related to only to the optimized process but also to the optimisation process itself and its consequences outside the targeted process).

Please add a short chapter within materials and methods or results analysing the connections between the four points illustrated in materials and methods.

Results

Tables 1 and 2 illustrates the construction of the archetypes of LCA goal and scope definitions. Actually, consequential LCA deals with not only global impact of a product but firstly on answering “what happens when the production of the product A increases/decreases of one unit?”. I think it is important to include also this aspect in the results (and discussion) section.

Discussion

About the cut-off argument, it seems that cut-off is an allocation procedure. I think it is better to explain that the cut-off deals also with the exclusion of some production phases if their contribution to the whole impact is under the cut-off criteria, in order to not mislead the reader.

About lines 274-279, authors stated that decision-making aim could be achieved both with ALCA and CLCA. I think it is questionable. Usually, the decision implies that a new method/process substitutes an old method/process, for example because the new one is more efficient. The consequence of a decision is a change in the production system itself (a marginal increase in the new method and a marginal decrease in the old one) and indirectly in other production systems connected to it. As ALCA is a photograph of the present and it does not include a “dynamic” view, ALCA is not the best method to assess the changes in direct and indirect the environmental impacts. Please modify this paragraph in order to have a complete view on the issue.

Specific comments

Abstract

Line 15: I suggest to split the sentence. “…are developed. These archetypes reflect…”

Introduction

Line 28: “guidelines providing divergent

Line 35: system expansion (as synonym of substitution) or system expansion as stated by Heijungs 2013? Read with the earlier and subsequent sentences, the line could be misinterpreted. Please clarify.

Materials and methods

None

Results

Line 176: Azapagic & Clift is reported with number 19 in the text and with number 17 in the references. Please check all the references’ number (for example Neugebauer et al., or Benoit Norris) throughout the manuscript

Table 1 Please check the setting of the table. I suggest to include the symbols (alfa, betta, gamma) in a new column.

Table 1 “the reason for carrying…” not bold

Discussion

Line 335. Please add a comma after “(SLCA)”

Conclusion

Lines 358-359. “…are developed, reflecting…”

References

Line 396. The title of the reference not in italics

Reviewer 2 Report

This is a very interesting study in the field of Life Cycle Assessment (LCA) via an illustrative example (a case study) where archetypes of goal and scope definitions are developed for consistent allocation. This topic is relevant for the wider community of Life Cycle Assessment (LCA) scientists, and for some people concerned with environmental and sustainability issues. The paper is well written, and properly referenced. However, I think the paper could be consolidated with some minor changes. Below I add my suggestions on how these improvements could be done.

Title: Here’s my suggestion for the title: “Archetypes of goal and scope definitions for consistent allocation in LCA – a case study”

This study is just a case study, as the authors state on page 2, lines 77–78: “We use the example of the recycling of rare earth elements from end-of-life fluorescent lamps”. This work is part of a work developed within the context of the PhD of the first author, which was funded by Solvay Company and the French National Association for Technical Research. So, the authors must be careful for nothing can be proved with an example, and this paper only presents an illustrative example (a case study). They can only refute aspects (provide a counterexample: if you can provide one example that is false, you have proven the entire conjecture to be false).

Page 13, lines 359–362: “It is demonstrated via an illustrative example that the question “what is the environmental impact of a product” can result in at least 15 different research questions requiring at least 5 different modeling methods.” Here again, nothing is proved via an example, and besides I don’t understand the segment “15 different research questions requiring at least 5 different modeling methods”, because in Table 3 in “Modeling specifics” are presented just 3 different modeling methods. The authors must explain and justify more convincingly these statements that appear several times (in the Abstract). These are topics that the reader can appreciate but they must be rigorously developed and justified: for example, regarding the systems boundaries of the foreground subsystem (Figure 1), the authors state: “processes operated by the company Solvay”), and the LCA study (Figure 3, Figure 4 and Figure 5).

I advise the authors to review data presentation. It is widely known that usefulness and reliability of the results from an LCA study depend on the quality of the data that is used as basis, i.e., the data that describes the processes included in the different parts of the product system. Therefore, it is important to define data requirements and data quality requirements. The authors must clearly define what data was collected (how and where the data was collected). The authors should also justify the precision, completeness and representativeness of the data, as well as the consistency and reproducibility of the methods used throughout the study.

Page 1 (Introduction), line 29: “ISO” should be “International Organization for Standardization (ISO)”. Keep in mind that the reader is not necessarily knowledgeable on the field addressed in the paper and so must be minimally elucidated in order to be able to read and understand the paper.

Page 11, line 227: “2. Mathematical equivalence”. I disagree with this statement. Do the authors know what equivalence in mathematics is? Rigorously speaking, a mathematical equivalence is not what the authors are stating and so they should remove the word “mathematical”.

I suggest adding information about the definition of the parameters of interest (and ) presented in Table 1. How are they estimated or measured? How do they vary? For example, is  related with “The production/treatment and/or ??? The consumption”?

Regarding formatting issues:

Table 3 must be divided in several tables because it is not possible to present it in a single page;

Figure 4 and Figure 5 override the page margins and must be shortened;

Page 12, lines 266 and 267 are not formatted;

References:

References 17, 24, 31 and 32 are not reported in the manuscript.
